**Freshening of Antarctic Intermediate Water in the South Atlantic Ocean in 2005 - 2014**
Wenjun Yao[a,*], Jiuxin Shi[a], Xiaolong Zhao[a]
[1]Key Lab of Physical Oceanography (Ocean University of China), Ministry of Education,
Qingdao 266100, Shandong, China
*Correspondence to:* E-mail address: wjimyao@gmail.com (Wenjun Yao)
**Abstract**
Basin-scale freshening of Antarctic Intermediate Water (AAIW) is reported to have
occurred in the South Atlantic Ocean during the period from 2005 to 2014, as shown by the
gridded monthly means Array for Real-time Geostrophic Oceanography (Argo) data. This
phenomenon was also revealed by two repeated transects along a section at 30° S, performed
during the World Ocean Circulation Experiment Hydrographic Program. Freshening of the
AAIW was compensated by a salinity increase of thermocline water, indicating a hydrological
cycle intensification. This was supported by the precipitation less evaporation change in the
Southern Hemisphere from 2000 to 2014. Freshwater input from atmosphere to ocean surface
increased in the subpolar high-precipitation region and *vice versa* in the subtropical
high-evaporation region. Against the background of hydrological cycle changes, a decrease in
the transport of Agulhas Leakage (AL) which was revealed by the simulated velocity field,
was proposed to be a contributor to the associated freshening of AAIW. Further calculation
showed that such decrease could account for approximately 53% of the observed freshening
(mean salinity reduction of about 0.012 over the AAIW layer). The estimated variability of
AL was inferred from a weakening of wind stress over the South Indian Ocean since the
beginning of the 2000s, which would facilitate freshwater input from the source region. The
mechanical analysis of wind data here was qualitative, but it is contended that this study
would be helpful to validate and test predictably coupled sea-air model simulations.
Keywords: Freshening; Antarctic Intermediate Water; South Atlantic; Agulhas Leakage;
Wind Stress

[a] Key Lab of Physical Oceanography (Ocean University of China), Ministry of Education, Qingdao 266100, Shandong, China

* Corresponding author at: Key Lab of Physical Oceanography (Ocean University of China), Ministry of Education, 238 Shongling Road, Laoshan District, Qingdao 266100, Shandong, China. Mob: +86 151 5421 9251. E-mail address: wjimyao@gmail.com (Wenjun Yao)

## 1 Introduction

Thermocline and intermediate waters play an important part in global overturning circulation by ventilating the subtropical gyres in different parts of the world oceans [*Sloyan and Rintoul*, 2001]. They also constitute the northern limb of the Southern Hemisphere supergyre [*Ridgway and Dunn*, 2007; *Speich et al.*, 2002].

Previous studies have addressed the variability of intermediate water. *Wong et al.* [2001] found that the intermediate water had freshened between the 1960s and the period 1985-94 in the Pacific Ocean. *Bindoff and McDougall* [2000] reported that there had been freshening of water between 500 and 1500 db from 1962 to 1987 along 32° S in the Indian Ocean. *Curry et al.* [2003] showed a salinity reduction on the isopycnal surface of intermediate water for the period 1950s - 1990s in the western Atlantic. The freshening variability can be traced back to the signature of water in the formation regions [*Church et al.*, 1991]. The freshening examples given above are in agreement with the enhancement in hydrological cycle, in which the wet (precipitation ($P$) > evaporation ($E$), $P$ dominance) subpolar regions have been getting wetter and vice versa for the dry ($E$ dominance) subtropical regions over the last 50 years [*Held and Soden*, 2006; *Skliris et al.*, 2014].

Antarctic Intermediate Water (AAIW) is characterized by a salinity minimum (core of AAIW) centered at the depths of 600 m and 1000 m (Fig. 1Fig. 1), which lies within potential density (reference to sea surface) range of $\sigma_0$ = 27.1 - 27.3 kg/m$^3$ [*Piola and Georgi*, 1982]. The AAIW is found from just north of the Subantarctic Front (SAF) [*Orsi et al.*, 1995] in the Southern Ocean and can be traced as far as 20° N [*Talley*, 1996]. It is generally accepted that the variability of AAIW is largely controlled by air-sea-ice interaction [*Close et al.*, 2013; *Naveira Garabato et al.*, 2009; *Santoso and England*, 2004], but the argument about its origin and formation process continues. For example, there is the circumpolar formation theory of AAIW along the SAF, through mixing with Antarctic Surface Water (AASW) along isopycnal [*Fetter et al.*, 2010; *Sverdrup et al.*, 1942]. Alternatively, it has been proposed that there is a local formation of AAIW in specific regions, as a bi-product of Subantarctic Mode

Water (SAMW) relating to deep convection [*McCartney*, 1982; *Piola and Georgi*, 1982]. The first standpoint states that the AAIW are primarily derived from entire subpolar sources, meanwhile the second one emphasizes the role that air-sea interaction plays in the oceans south of South America.

In the South Atlantic, AAIW constitutes the return branch of the Meridional Overturning Circulation (MOC) [*Donners and Drijfhout*, 2004; *Speich et al.*, 2007; *Talley*, 2013]. As an open ocean basin, the South Atlantic is fed by two different sources of AAIW [*Sun and Watts*, 2002]. The first is younger, fresher and has a lower apparent oxygen utilization (AOU) and originates from the Southeast Pacific [*McCartney*, 1977; *Talley*, 1996] and the winter waters west of Antarctic Peninsula [*Naveira Garabato et al.*, 2009; *Santoso and England*, 2004]. These source regions of AAIW are mostly dominated by the net surface freshwater flux from atmosphere to ocean ($P>E$), which facilitates the freshening of AAIW with time. The second is the older, saltier and higher AOU AAIW which comes from the Indian Ocean transported by the Agulhas Leakage (AL) as Agulhas rings (Fig. 2Fig. 2). The mixture of the above two types of AAIW can lead to a transition of hydrographic properties across the subtropical South Atlantic [*Boebel et al.*, 1997].

The influence of AL on variability of AAIW in the South Atlantic has been demonstrated to be substantial [*Hummels et al.*, 2015; *Schmidtko and Johnson*, 2012], as 50 - 60% of the Atlantic AAIW originates from the Indian Ocean [*Gordon et al.*, 1992; *McCarthy et al.*, 2012], with increased (decreased) transport of AL relating to salinification (freshening) of AAIW. AL has apparently increased during the period from 1950s to the early 2000s [*Durgadoo et al.*, 2013; *Lübbecke et al.*, 2015], but there have been no studies addressing the influence of AL on the AAIW in South Atlantic since 2000.

With the instigation of the Array for Real-time Geostrophic Oceanography (Argo) program, *in-situ* hydrographic observation has tremendously expanded since 2003 [*Roemmich et al.*, 2015], particularly in the Southern Ocean (SO) where historical data are sparse and intermittent. This decreases the uncertainty of estimates for the research on both seasonal and decadal variations of subsurface and intermediate waters.

The present work reported the freshening of AAIW in the South Atlantic over the
preceding decade (2005 - 2014) using gridded monthly data based on Argo data. Against the
background of an enhanced hydrological cycle, decreased transport of AL contributed to such
freshening and may be driven by a weakening of wind stress in the South Indian Ocean
during the same period.
**2    Data and methods**
Based on individual temperature ($T$) and salinity ($S$) profiles from Argo, International
Pacific Research Centre (IPRC) gridded monthly means data for the period 2005 and 2014 are
produced using variational interpolation. The IPRC data have 27 levels from 0 to 2000 m
depth vertically, nominal 1°×1° grid globally and monthly temporal resolution
(http://apdrc.soest.hawaii.edu/projects/Argo/data/gridded/On_standard_levels/index-1.html).
To reduce the error from low vertical resolution of data when computing the hydrographic
values on isopycnal surface, $T$ and $S$ profiles were first interpolated onto 1 m depth intervals
vertically using spline method in the intermediate water depth, and linear method in the
thermocline depth. Because the IPRC data were interpolated from randomly distributed Argo
profiles, it is necessary to demonstrate the robust nature of their signals by comparing with
the other Argo gridded products. As a result, the Japan Agency of Marine-Earth Science and
Technology (JAMSTEC, [*Hosoda et al.*, 2008]) $T$ and $S$ data from 2005 to 2014 with 1°
longitude and 1° latitude resolution were also collected for comparison and verification. The
number of Argo profiles is rapidly increasing year by year, and part of their distribution has
been outlined in previous studies, *inter alia Hosoda et al.* [2008] and *Roemmich et al.* [2015].
Two hydrographic cruises of repeated transects along 30° S were conducted in the World
Ocean       Circulation       Experiment       (WOCE)       Hydrographic       Program
(http://www.nodc.noaa.gov/woce/wdiu/diu_summaries/whp/index.htm). Their locations are
presented in Fig. 2Fig. 2. The first transect consisted of 72 stations in 2003 by the R/V Mirai
(Japan, [*Kawano et al.*, 2004]), the second was in 2011 with 81 stations sampled from the
Ronald H. Brown (United States, [*Feely et al.*, 2011]). These two transects were not only
performed in almost the repeated positions in the subtropical South Atlantic, but also
conducted in the same season (November and October respectively). Furthermore, the time
interval between the two sections from Nov 2003 to Oct 2011 is very similar to the IPRC
covered period (Jan 2005 - Dec 2014) and can therefore be used to validate those results.
To smooth out some of the higher frequency variability (i.e. mesoscale eddies and
internal waves), the investigation of halocline variation should be along neutral density
surfaces [*McCarthy et al.*, 2011; *McDougall*, 1987]. The layer of AAIW is defined using
neutral density ($\gamma^n$, unit: kg/m$^3$) [*Jackett and McDougall*, 1997] instead of potential density,
with the upper and lower boundaries being $27.1\gamma^n$ and $27.6\gamma^n$ [*Goes et al.*, 2014], respectively.
Monthly 10 m wind fields between years 1980 and 2014 from the ERA-Interim archive
at the European Centre for Medium Range Weather Forecasts (ECMWF)
(http://apps.ecmwf.int/datasets/data/interim-full-daily/levtype=sfc/) were used to investigate
the decadal variability of wind stress (WS) over the South Indian Ocean. Another reanalysis
wind product of National Centers for Environmental Prediction-Department of Energy
Atmospheric Model Intercomparison Project reanalysis 2 (NCEP2,
http://www.esrl.noaa.gov/psd/data/gridded/data.ncep.reanalysis2.html) was also used for the
period 1980-2014. Additionally, the satellite-derived wind products of Quick Scatterometer
(QuikSCAT) for 2000-2007 and Advanced Scatterometer (ASCAT) for 2008-2014 (both in
ftp://ftp.ifremer.fr/ifremer/cersat/products/gridded/MWF/L3/) were used to compare and
verify the decadal variability of WS revealed by the ERA-Interim wind product. In this work,
the WS over open ocean was calculated from 10 m wind field data using the equation adopted
in *Trenberth et al.* [1989].
Reanalysis data including precipitation (*P*) and evaporation (*E*) from the ERA-Interim
were used to reveal the freshwater input from the atmosphere to ocean surface in the
preceding decade.
The Simple Ocean Data Assimilation version 3.3.1 (SODA3.3.1,
http://www.atmos.umd.edu/~ocean/), which is forced by the Modern-Era Retrospective
analysis for Research and Applications Version 2 (MERRA2), spans the 36-year period
1980-2015 ([*Carton et al.*, 2016]). The global simulated velocity field at specified depths
provided by SODA make it available to evaluate the transport of AL.
**3    Freshening of Antarctic Intermediate Water**
**3.1    Freshening observed from Argo gridded products**
The Argo gridded products provide a globally distributed and continuous time series of $T$
and $S$ profiles down to 2000 m ocean depth. The present work focused on the AAIW in the
South Atlantic Basin (Fig. 2~~Fig. 2~~, Region A), which encompasses most of the subtropical
gyre and a part of the tropical regimes [*Boebel et al.*, 1997; *Talley*, 1996]. Computed from the
Argo gridded data of IPRC, the biennial mean of $\theta$-$S$ diagram (Fig. 3~~Fig. 3~~a) clearly shows
that the AAIW has experienced a process of progressive basin-scale freshening during the
period from Jan 2005 to Dec 2014. The linear trend of salinity (Fig. 3~~Fig. 3~~b) further reveals
that the freshening takes up most of the AAIW layer but with a little salinification in the
deeper part of it. Except around the $27.42\gamma^n$ neutral density surface, the AAIW variation is
significant at 95% confidence level, using the $F$-test criteria. In comparison with Fig. 3~~Fig. 3~~a,
it was found that the cut-off point of transformation from salinity decrease to increase is near
the salinity minimum. Above the salinity minimum, the shift of $\theta$-$S$ trends towards cooler and
fresher values along density surface and seem to be a response to the warming and freshening
of surface waters where AAIW ventilates. Such thermohaline change has also been found in
the Pacific and Indian oceans over a different time period [*Wong et al.*, 1999]. *Church et al.*
[1991] and *Bindoff and Mcdougall* [1994] have researched the counterintuitive cooling of
AAIW temperature induced by warming of surface water. They showed that a warming parcel
at mixed layer would subduct further equatorward, which would lead the $\theta$-$S$ curve to become
cooler and fresher at a given density. The salinity decrease of core of AAIW indicates that
such change can only be induced by freshwater input from the source region, as mixing with
more saline surrounding waters cannot give rise to a salt loss in the salinity minimum layer.
To demonstrate the robustness of AAIW variations revealed by the IPRC data, re-plots
of Fig. 3Fig. 3a-b using another Argo gridded product, the JAMSTEC, were also shown for
comparison (see Supplementary 1, only the AAIW layer shown). Not only the same variation
along density surfaces in the AAIW layer was found, but also a freshening of the salinity
minimum. Both the isoneutral salinity increases of IPRC and JAMSTEC data below the
salinity minimum are quite small. The main discrepancy between them is that the salinity
reduction in the JAMSTEC data is somewhat (a mean of 0.006 between $27.1\gamma^{n}$ and $27.6\gamma^{n}$)
less than IPRC and at a higher 95% confidence level.
The freshwater gain for the basin-scale salinity decrease of AAIW (mean salinity
difference of 0.012 between $27.1\gamma^{n}$ and $27.6\gamma^{n}$ over a mean water mass thickness of 500 m)
was estimated at 17mm yr$^{-1}$ in its source region (Assuming the case that the South Atlantic
only experienced freshwater input and nothing changed, thus the relationship between the
salinity in 2005 and 2014 in unit area was roughly $S_{2005}*500 = S_{2014}*(500+\Delta d)$. Here $S_{2005} =$
$S_{2014}+0.012$ and $\Delta d$ is the freshwater gain during the covered period). However, the
depth-integrated salinity change over the water column (between $26.6\gamma^{n}$ and $27.6\gamma^{n}$) was
0.0014, since a salinity increase of thermocline water balances the observed freshening of
AAIW. This salinity budget implies contemporary hydrological cycle intensification in the
southern hemisphere, which is illustrated by the $P$ minus $E$ change from 2000 to 2014, with
$P$-$E$ increasing in the subpolar region and *vice versa* in the subtropical region (Fig. 4Fig. 4a).
In these cases, the thermocline (intermediate) water that ventilates in the high-evaporation
(precipitation) subtropical (subpolar) regions gets more saline (freshened), as shown by the
hydrographic observations (Fig. 3Fig. 3b).
Against the background of hydrological cycle augmentation, the annual freshwater input
in the AAIW ventilation region during the freshening period increased by 0.02 mm day$^{-1}$,
about 17% of the $P$-$E$ in 2005 (Fig. 4Fig. 4b). It is considered that the significant $P$-$E$ increase
began around 2003 (Fig. 4Fig. 4b, 5-yr running mean line), which means the observed
freshened AAIW could be traced back to 2003. Though it was not possible to compute the
direct freshwater input to the South Atlantic Basin in this study, the Argo era freshening of
AAIW is qualitatively consistent with the freshwater gain in its source region.

## 3.2 Freshening in the quasi-synchronous WOCE CTD observations

Here two synoptic transatlantic sections from WOCE hydrographic program were used to explore the decadal freshening signal identified in the above subsection. Similar to Fig. 3~~Fig. 3~~a, sectional mean $\theta$-$S$ diagram (Fig. 5~~Fig. 5~~a) displays the same shift of thermohaline values, including freshening of the salinity minimum, salinity reduction in the upper AAIW layer and *vice versa* in the lower layer. Compared to the $\theta$-$S$ curves of IPRC data (Fig. 3~~Fig. 3~~a), the curves of WOCE (Fig. 5~~Fig. 5~~a) seem to be, in general, cooler $\theta$ and fresher $S$. It is suggested that this is because the IPRC mean is weighted towards the warmer and saltier waters in the north.

Unlike the Argo gridded product which has continuous time series of $T$ and $S$ data, there are only two sections in the WOCE observations. Instead of calculating the linear trend of salinity (as was done with the IPRC data), the difference of salinity observed in 2003 and 2011 was estimated (Fig. 5~~Fig. 5~~b). The light grey shading denotes the 95% confidence interval using simple $t$-test criteria and having consider the number of degrees of freedom. Above the salinity minimum, the freshening of AAIW revealed by the IPRC and the WOCE data are quite similar, with the maximum appearing near $27.2\gamma^n$. Because the last WOCE observation terminated in 2011 and the salinity reduction would continue at least up to 2014 as displayed in Fig. 3~~Fig. 3~~a, the magnitude of the freshening in WOCE (Fig. 5~~Fig. 5~~b) is smaller than IPRC (Fig. 3~~Fig. 3~~b). In the water layer below the salinity minimum (around $27.41\gamma^n$), the salinity increase shown in the WOCE data is relatively large (Fig. 5~~Fig. 5~~b). This is thought to be because the salinity rise reached its maximum around 2011, which is shown in the time series of basinwide averaged salinity on $27.45\gamma^n$ and $27.55\gamma^n$ density surfaces (see Supplementary 2).

For the salinification of thermocline water, there is a large discrepancy between IPRC and WOCE data, on neutral density surfaces $26.6$-$26.7\gamma^n$ (Fig. 5~~Fig. 5~~b). It is considered that this would not affect the salinity budget over the water column (Fig. 5~~Fig. 5~~b), given that the salt gain of thermocline water would balance the observed freshened AAIW. In conclusion, the general trend and consistency of the detail therein of the salinity change over the last

ten-year time period revealed by the IPRC and the WOCE data, leads us to state that the
freshening of AAIW is a robust finding and valid.

## 4    Decrease of Agulhas Leakage transport

AAIW in the South Atlantic is largely influenced by the AL through the intermittent
pinching off of Agulhas rings (Fig. 2~~Fig. 2~~) [*Beal et al.*, 2011], transferring salty thermocline
and intermediate water from the Indian Ocean to the South Atlantic [*De Ruijter et al.*, 1999].
The above discussion suggests that the freshening of AAIW was induced by the input of
freshwater from the source regions, which are consisted of the southeast Pacific Ocean and
the circumpolarly subpolar oceans (see introduction). As a result, if the transport of more
saline water from the Indian Ocean decreased, it would promote the effect of this freshwater
increase. In this section, the decrease of AL transport was evaluated by depth integration of
velocity field and further demonstrated by using an indirect indicator.

### 4.1    Evaluation from SODA velocity

In the study of modeling, it is widely acceptable to quantify the leakage follows a
Lagrangian approach [*Biastoch et al.*, 2009; *van Sebille et al.*, 2009]. Here a simplified
strategy was employed to compute the leakage by integrating the velocity within AAIW layer
(approximately between 610 and 1150m, according to Fig. 1~~Fig. 1~~), which was demonstrated
to result to a similar quantification to the Lagrangian one [*Le Bars et al.*, 2014]. The depth
integration is along the Goodhope section (green line in Fig. 2~~Fig. 2~~), using the
cross-component velocity. Note that the leakage calculation is from the continent to the zero
line of barotropic streamfunction, which is the separation of the Agulhas regime and the
Antarctic Circumpolar Current [*Biastoch et al.*, 2015].
Before showing the transport computed from the SODA velocity data, it is necessary to
verify that the SODA hydrographic data could reveal the same freshening of AAIW as the
other dataset done. And in consequence, the AAIW in the South Atlantic was also shown to
have freshened during period 2005-2014, though with relatively small magnitude
(Supplementary 3). Yearly leakage computation within AAIW layer was employed for the
period 2000-2015 (Fig. 6Fig. 6). It shows that the leakage in the early 2010s is smaller than
that in the middle and post 2000s, forming a decreased trend in a nearly ten-year period. This
estimation of leakage seems to be consistent with the below indirect estimation of AL
transport.
The following calculation is to simply estimate the contribution of the above AL
transport change to our observed freshening. As shown by Fig. 6Fig. 6, the decreased rate of
AL transport could be taken to be 2 Sv in a ten-year time period. And assuming that this rate
increased year by year in the study period (i.e., 0.2 Sv in the first year, 0.4 Sv in the second
year, and so on.). According to *Sun and Watts* [2002], here we take the salinity difference of
$\Delta S$=0.1 between the South Indian and the South Atlantic in the AAIW layers. The other
parameters, including total seconds in a year, water thickness of the AAIW layer, the area of
Region A, are taken to be $\Delta t$=365×24×3600s, $\Delta d$=500 m and $\Delta s_A$=1.09×10$^{13}$ m$^2$, respectively.
Therefore, the salinity decrease from 2005 to 2014 induced by the change of AL transport,
should be (0.2+0.4+…+2)×10$^6$×$\Delta t$×$\Delta S$/($\Delta s_A$×$\Delta d$). As a result, a 0.0064 of salinity reduction
was induced, which could account for approximately 53.0% of the observed freshening
revealed by the IPRC data. Though our estimation here was quite roughly, through which we
could be safety to state that, in the years 2005-2014, the AL behaved to significantly influence
the salinity change in the South Atlantic Ocean within the AAIW layers.
**4.2    Weakening of the westerlies in the South Indian Ocean**
Continuous measurements of the AL transport have never been realized so far. The
earlier study suggested that an increased AL transport correlates well with a poleward shift of
westerlies [*Beal et al.*, 2011]. However, after using reanalysis and climate models, *Swart and*
*Fyfe* [2012] argued that the strengthening of Southern Hemisphere surface westerlies has
occurred without major transgressions in its latitudinal position over the period 1979-2010,
during which period the AL has largely increased [*Biastoch et al.*, 2009]. A more recent study
of *Durgadoo et al.* [2013] showed that the increase of AL is concomitant with equatorward
rather than poleward shift of westerlies in their simulation cases. They also concluded that the

intensity of westerlies is predominantly responsible in controlling this Indian-Atlantic transport. Many relevant studies agreed on this relationship, that the enhancement of westerlies intensity is related to the increase of AL [*Goes et al.*, 2014; *Lee et al.*, 2011; *Loveday et al.*, 2015].

The AL corresponds most significantly to westerlies strength averaged over the Indian Ocean in contrast to that averaged circumpolarly or locally [*Durgadoo et al.*, 2013]. According to the work of *Durgadoo et al.* [2013], zonally averaged WS was calculated from the wind product of ERA-Interim over the Indian Ocean (20-110° E) for every 5-yr period since 1980 (Fig. 7~~Fig. 7~~a and d). Previous studies [*Lee et al.*, 2011; *Loveday et al.*, 2015] have found that the WS has considerably increased from 1980s to the beginning of 2000s (Fig. 7~~Fig. 7~~d), consistent with the contemporary increase of AL transport. Though there are oscillations during 1990s, the WS reached its peak around the years 2000-2004 (Fig. 7~~Fig. 7~~d), then began to decline. It can be concluded that the WS has weakened for period 2000 – 2014 (Fig. 7~~Fig. 7~~d), which implies a concurrent decrease of AL transport.

In addition to the ERA-Interim wind data, we have further checked the zonally averaged WS over the Indian Ocean (20-110° E), using another reanalysis product of NCEP2 (Fig. 7~~Fig. 7~~b and e) and the combined QuikSCAT-ASCAT (Fig. 7~~Fig. 7~~c and f) satellite-derived wind products. The three zonally averaged WS agree that during the period 2000-2014, the westerlies reached a peak in the years 2000-2004, and then progressively subsided through 2005-2009 to 2010-2014. The process of gradual decline of WS is most pronounced in the NCEP2 data. It is noteworthy that none of the three products show a significant meridional shift of the latitude of maximum WS from 2000 to 2014, in corroboration with the conclusion of *Swart and Fyfe* [2012].

**4.3   Evidence from other works**

Many efforts have been made to estimate AL transport, especially using model simulations [*Lübbecke et al.*, 2015; *Loveday et al.*, 2015]. In recent years, *Le Bars et al.* [2014] provided the time series of AL transport over the satellite altimeter era, computed from absolute dynamic topography data, which can show the decadal variation of AL present. In

their result (Figure 8 in *Le Bars et al.* [2014]), the anomalies of AL from satellite altimetry
reached a peak around 2003 (annual average), and then began to subside, apart from a
mid-2011 increase. In addition, their negative trend of AL (Figure 9 in *Le Bars et al.* [2014])
over the period from Oct 1992 to Dec 2012 indicates that the transport was reduced during the
2000s in contrast to the 1990s. Another study by *Biastoch et al.* [2015] should be of help in
the present discussion. Though the time series of AL obtained from models didn't show a
distinct decline of AL transport in the last decade, which seems partly due to the data filter
applied and the end of time series in 2010 (Figure 4 in *Biastoch et al.* [2015]), it displays a
maximum of salt transport around 2000 (Figure 5 in *Biastoch et al.* [2015]). This peak and the
subsequent decline of salt transport are consistent with the freshening of AAIW over the
similar time period considered here.
Thus, in addition to the freshwater input that gave rise to the salt loss of the AAIW in the
South Atlantic Ocean, reduced transport of AL or salt would further enhance this signal.
Unfortunately, the analyses of the contribution from both the source region and AL were
qualitative or roughly quantitative. Future work should be focused on quantification of each
factor based on model simulations.

## 5    Conclusions and discussions

The analysis of IPRC gridded data shows that the AAIW in the South Atlantic has
experienced basin-scale freshening for the period from Jan 2005 to Dec 2014 (Fig. 3~~Fig. 3~~a
and b), with freshwater input estimated at 17 mm $yr^{-1}$ in its source region. Two transects of
WOCE hydrographic program observed in 2003 and 2011 also reveal the above variation of
AAIW in the last decade (Fig. 3~~Fig. 3~~c and d).
This freshening in the intermediate water layer was thought to be compensated by
increased salinity in shallower thermocline water, indicating a contemporary intensification of
hydrological cycle (Fig. 3~~Fig. 3~~b and Fig. 5~~Fig. 5~~b). In this case the freshwater input from
atmosphere to ocean surface increased in the subpolar high-precipitation region and *vice*

*versa* in the subtropical high-evaporation region (Fig. 4Fig. 4a). Over the last ten-year time period, significant freshwater gain began around 2003 (Fig. 4Fig. 4b), suggesting the observed freshened AAIW could be traced back to this time.

Against the background of hydrological cycle intensification, the decrease of AL transport was proposed to contribute to the freshening of AAIW in the South Atlantic, associated with a weakening of westerlies over the South Indian Ocean. This decrease was revealed by the leakage evaluation along the GoodHope section. The mechanical analysis shows that the WS over the South Indian Ocean reached its peak around 2000-2004 and began to subside through 2005-2009 to 2010-2014 (Fig. 7Fig. 7), reversing its increasing phase from 1950s to the beginning of 2000s, during which period the AL had increased [*Durgadoo et al.*, 2013; *Lübbecke et al.*, 2015]. This indirectly estimated variability of AL is consistent with other studies covering a similar period [*Biastoch et al.*, 2015; *Le Bars et al.*, 2014]. As the AAIW carried by the AL is more saline relative to its counterpart in the South Atlantic Ocean, its decrease would promote the effect of freshwater input from the source region. Our estimation further suggested that such induced freshwater input by AL could account for approximately 53% of the observed freshening.

May someone would ask if there are any other sources that could significantly affect the AAIW in the South Atlantic Ocean, for example the Southeast Pacific (see Introduction). To clarify this question, we displayed the EOF1 pattern and its time series (called the principal components) of salinity on the $27.36\gamma^{n}$ (around $27.2\sigma_0$) surface (Fig. 8Fig. 8) in the Southern Hemisphere, which explain the largest variance of 55.4%. It shows that in 2000-2014, the most significant salinity reduction appeared in the South Indian Ocean, especially in the region of Agulhas Current System. It also displays that compared to the West Atlantic, the East Atlantic experienced the major salinity reduction, whose intermediate water is largely fed by its counterpart in the South Indian. In addition to the above salinity change distribution, we also noted that the salinity decrease in the Southeast Pacific was quite less than that in the South Indian and the South Atlantic. Therefore, it implies that the Southeast Pacific did not play an important role at least in our observed AAIW freshening.

The purpose of this work is to reveal the decadal freshening of AAIW in the South
Atlantic Ocean over the last ten-year time period, and suggest the related contributing
mechanism. Future work should be focused on the quantification of these two contributors
through modelling simulation, and the influence they have on the world ocean circulation.

Acknowledgements
This study is supported by the Chinese Polar Environment Comprehensive Investigation
and Assessment Programs (Grant nos. CHINARE-04-04, CHINARE-04-01).

Captions of Figures

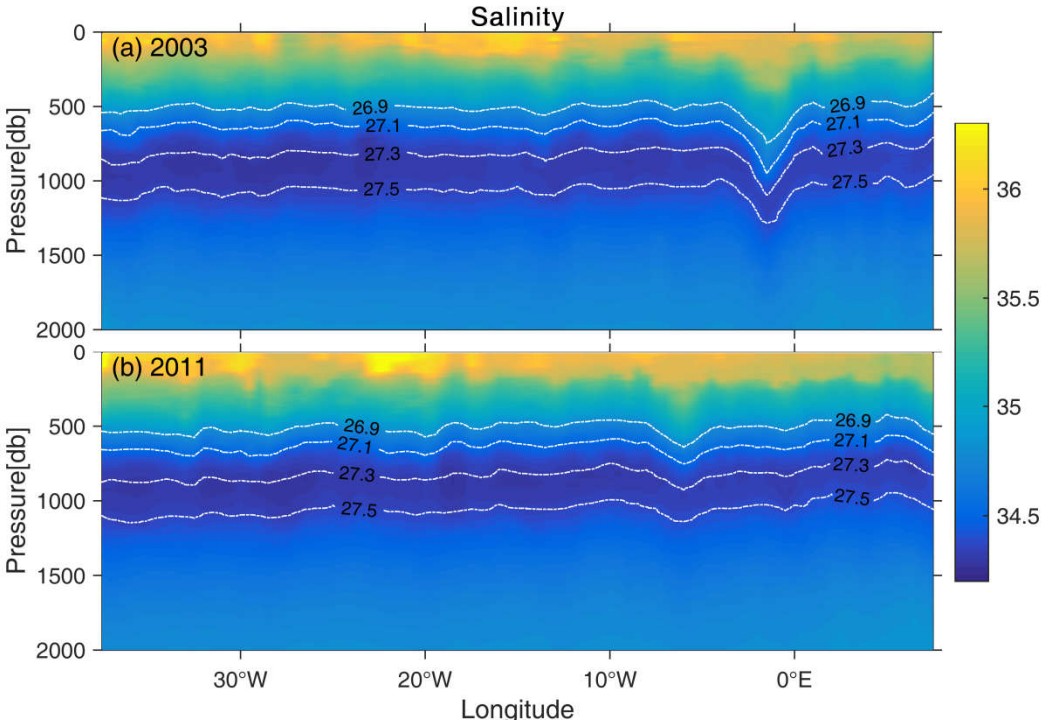


Fig. 1 WOCE salinity sections along 30° S in the South Atlantic Ocean (positions shown in
Fig. 2~~Fig. 2~~) observed in (a) 2003 and (b) 2011. Overlaid white solid-dotted lines are $\gamma^n$
surfaces ranging from 26.9 to 27.5 kg/m$^3$, with 0.2 kg/m$^3$ interval.

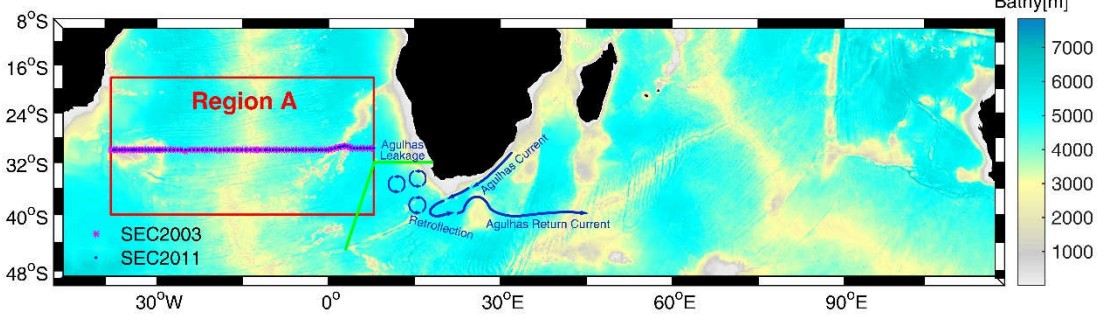

Fig. 2 Bathymetry of the South Indian-Atlantic oceans. Color shading is ocean depth. Red box delineates the area for the basinwide average of gridded data (hereafter refers to Region A). The green line shows the GoodHope section which is used to calculate the leakage transport to the South Atlantic. Magenta stars represent transatlantic CTD stations measured in 2003, meanwhile blue dots in 2011. The Agulhas Current, Retroflection, Agulhas Return Current and Agulhas Leakage (as eddies) are also shown and ticked.

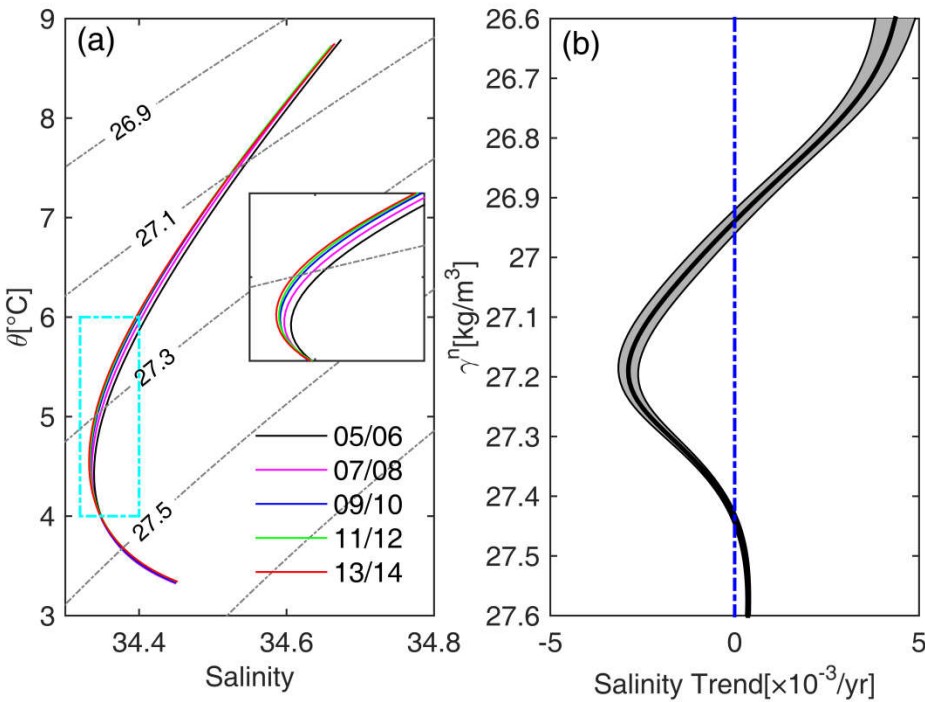

Fig. 3 (a) Biennial mean $\theta$-$S$ diagram averaged over Region A for IPRC data with $\gamma^n$ surfaces superimposed (grey solid-dotted lines). The inserted figure is the magnification of the area delineated by cyan solid-dotted box. The corresponding time for each $\theta$-$S$ curve is listed in their bottom-right corner (i.e. 05/06 for 2005-2006). (b) Salinity trend along $\gamma^n$ surfaces for period Jan 2005 – Dec 2014 is displayed by the thick black line, and the 95% confidence intervals ($F$-test) are represented by the light grey shadings, calculated from IPRC data.

388

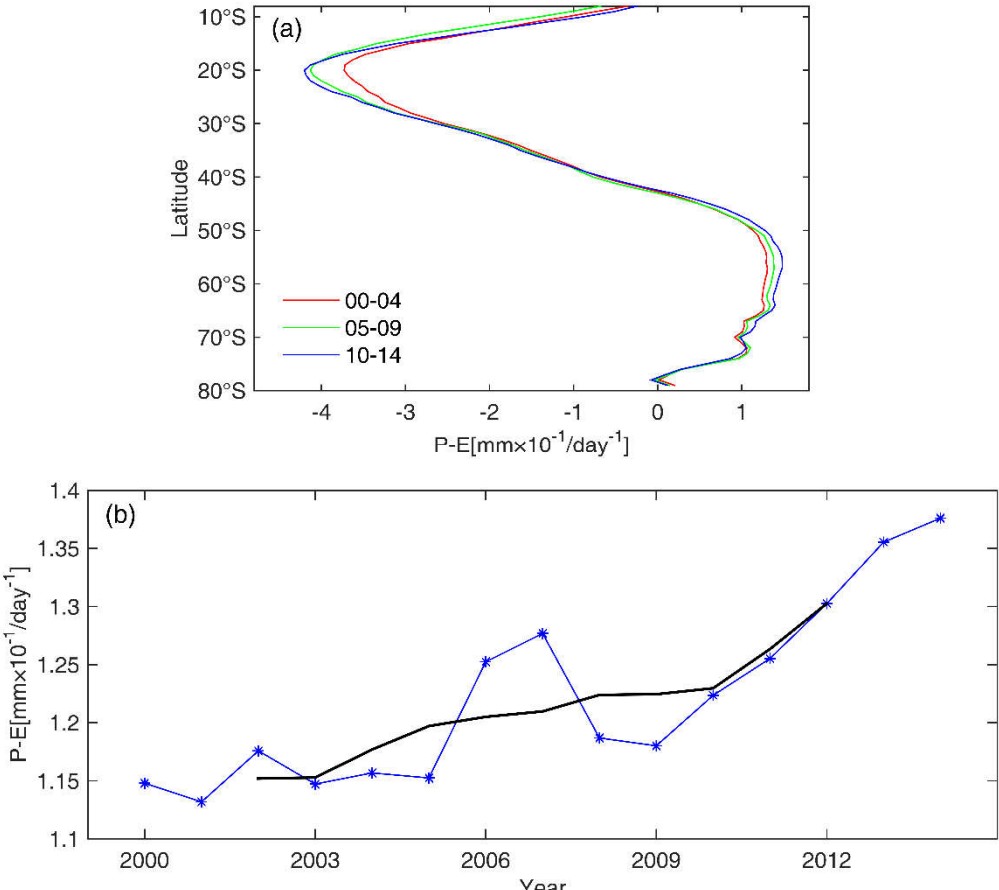

389

Fig. 4 Calculated from ERA-Interim precipitation and evaporation data: (a) Zonally mean (ocean areas only) of annually *P-E* (freshwater input, mm day$^{-1}$), each line represents a 5-yr averaged result. The corresponding time period (i.e. 00-04 for 2000-2004) is listed in the bottom-left corner. (b) Time series of annually *P-E* averaged over the oceans in 45-65° S, 0-360° E band from 2000 to 2014 (blue star), and its 5-yr running mean (black).


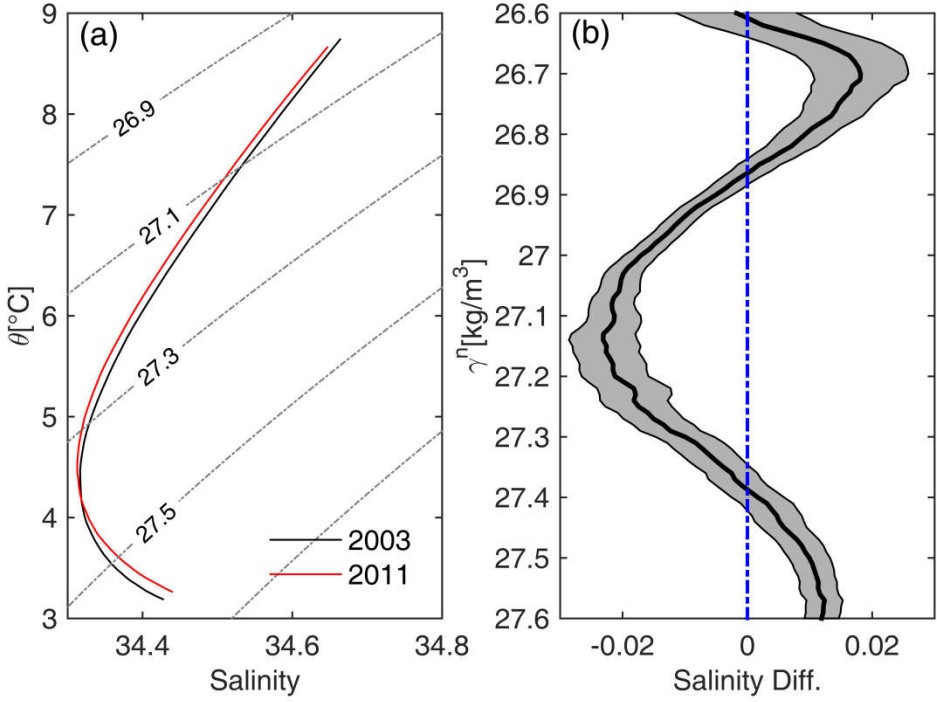

Fig. 5 (a) The same as Fig. 3Fig. 3a but for sectional mean of WOCE hydrographic casts. The
corresponding year for each $\theta$-$S$ curve is listed in their bottom-right corner. (b) Sectional
mean differences (thick black line) of WOCE hydrographic data along $\gamma^n$ and their 95%
confidence intervals (grey shadings, $t$-test).

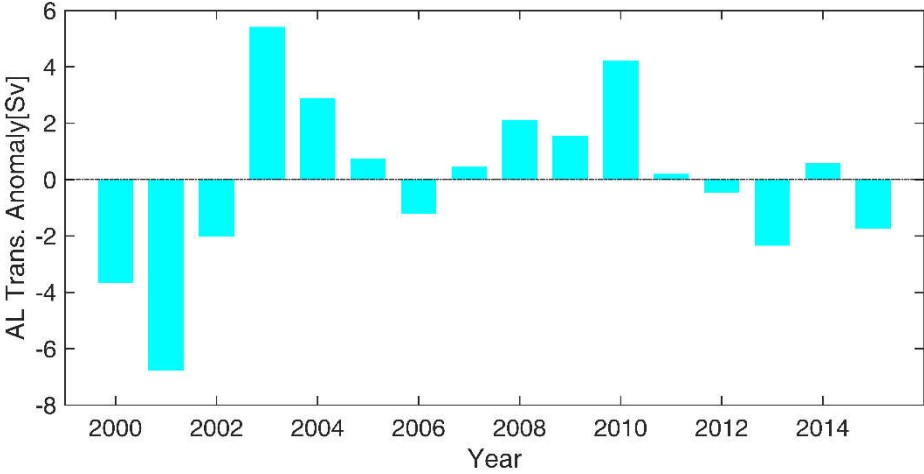

Fig. 6 Computation of Agulhas Leakage transport anomaly from the SODA velocity field
along the Goodhoop Line. Note that the depth integration is only for the AAIW layer.



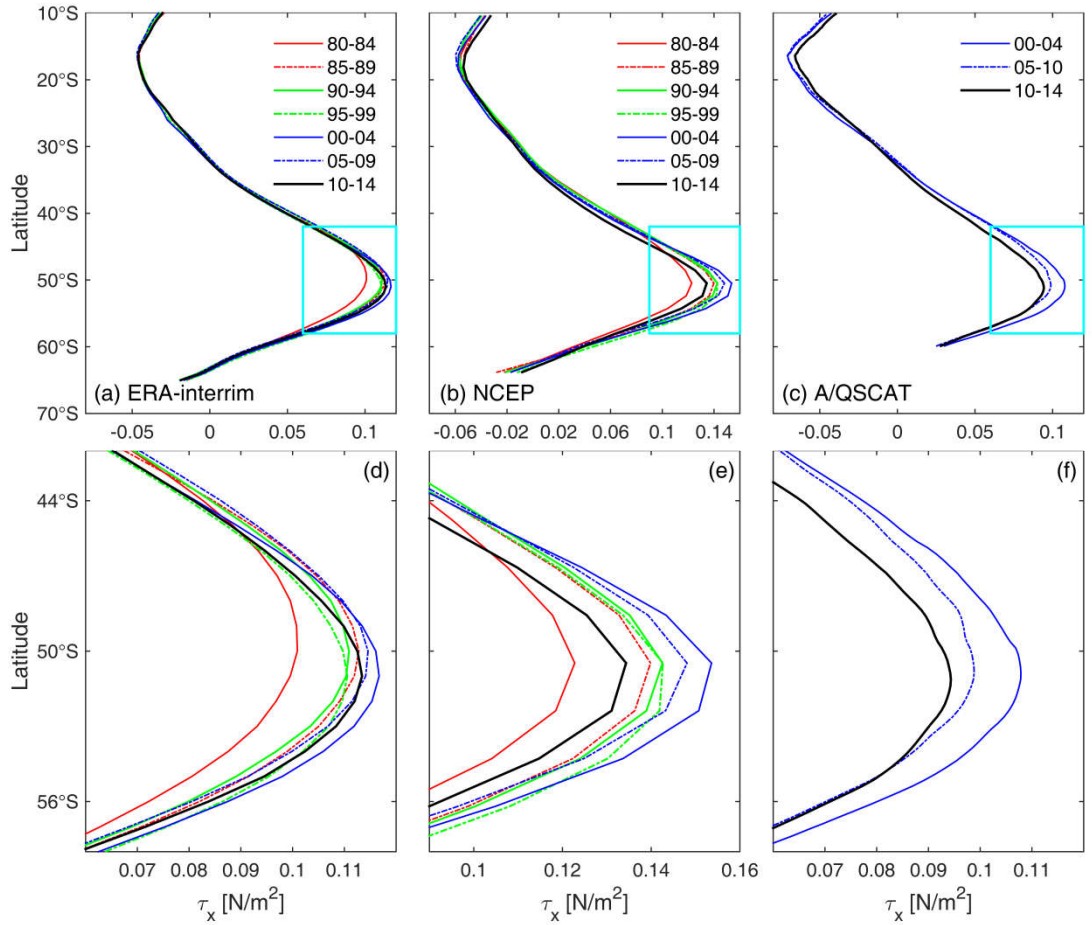

Fig. 7 Zonally averaged wind stress calculated from the wind product of (a) ERA-Interim, (b) NCEP2 and (c) ASCAT/QSCAT over the Indian Ocean (20° E-110° E) for different periods (i.e. 80-84 for Jan 1980 - Dec 1984; 00-04 for Jan 2000 - Dec 2004) listed in the top-right corners. (d), (e) and (f) are the magnification of cyan boxes in (a), (b) and (c), respectively.

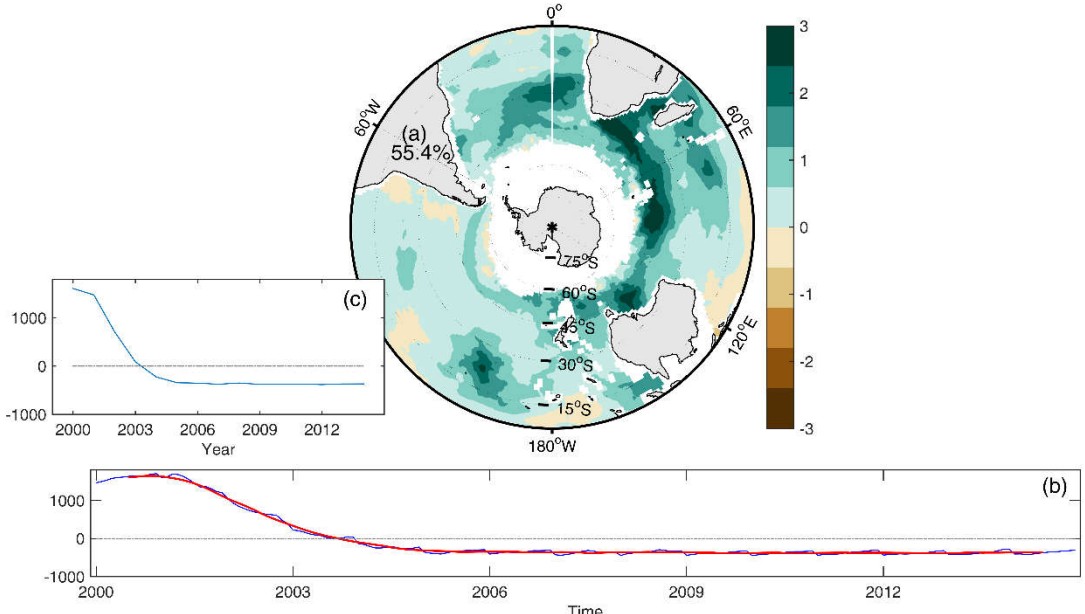

Fig. 8 (a) Pattern and (b) time series (blue: monthly, red: 13-month smooth) of EOF1 of salinity on $27.36\gamma^n$ surface. (c) Yearly mean time series of EOF1. Calculated from SODA data.

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
