# Peer review of "Freshening of Antarctic Intermediate Water in the South Atlantic Ocean in 2005 - 2014"

_Ocean Science, 2016_

## Referee Comment (RC1) · HN Waldron (Referee) · 15 Sep 2016

I have noted the comments of the Topic Editor that the paper is publishable from a scientific point of view in Ocean Science but that the standard of written english is poor. I agree. The paper uses elementary statistical methods to reach its conclusions and I advise the authors to avoid being definitive in their findings. Following the general comments of the Topic Editor I devoted my effort into improving the writing. To this end I converted the paper from.pdf to ms word and have made extensive edits in "track changes." I shall attempt to attach the file to this website submission but note that pdf and jpg files are specified. Alternatively, I will send it by email to Copernicus. I have already sent a copy to the Topic Editor.
* * *

---

## Referee Comment (RC2) · Anonymous Referee #1 · 15 Sep 2016

Edits in "Track Changes.

**Freshening of Antarctic Intermediate Water in the South Atlantic Ocean in 2005 -2014**

Wenjun Yao1 , Jiuxin Shi1

1Key Lab of Physical Oceanography (Ocean University of China), Ministry of Education,

Qingdao 266100, Shandong, China

Correspondence to: E-mail address: wjimyao@gmail.com (Wenjun Yao)

Abstract

Basin-scale freshening of Antarctic Intermediate Water (AAIW) is reported to have

 occurred in the South Atlantic Ocean during the period from 2005 to 2014, as shown by the gridded monthly means Argo (Array for Real-time Geostrophic Oceanography) data.

 This phenomenon was also revealed by two  repeated transects along a section  at 30°

S,  performed during the World Ocean Circulation Experiment Hydrographic Program. Freshening of the

AAIW was compensated by a salinity increase of thermocline water, indicating a hydrological cycle intensification. This was  supported by the precipitation less evaporation change in the Southern Hemisphere from 2000 to 2014, Freshwater input from atmosphere to ocean surface  increased in the subpolar, high-precipitation region and *vice versa* in the subtropical high-evaporation region. Against the background of hydrological cycle changes,  a decrease in the transport of Agulhas Leakage (AL) was proposed to be a contributor to the associated freshening of AAIW. This indirectly estimated variability of AL, was inferred from a weakening of wind stress over the South Indian Ocean since the beginning of the 2000s, which would facilitates  freshwater input from the source region and partly contributes to the observed  freshening of AAIW. The mechanical  analyses used in this study areis qualitative, but it is contended that this study  would be helpful to validate and test predictably coupled sea-air model simulations.

Keywords: Freshening; Antarctic Intermediate Water; South Atlantic; Agulhas Leakage;

Wind Stress

1. Introduction

Thermocline and intermediate waters play an important part in global overturning circulation by ventilating the subtropical gyres in different parts of the world oceans [Sloyan and Rintoul, 2001]. They also constitute the northern limb of the Southern

Hemisphere supergyre [Ridgway and Dunn, 2007; Speich et al., 2002].

Previous studies have addressed the variability of intermediate water. Wong et al. [2001]

found that the intermediate water had freshened  between the 1960s  and the period 1985-94 in the Pacific

Ocean Bindoff and McDougall [2000] reported that there had been a freshening of water between 500 and 1500 db from 1962 to 1987 along 32° S in the Indian Ocean, and Curry et al.

[2003] showed a salinity reduction on the isopycnal surface of intermediate water for the period 1950s -1990s in the western Atlantic. The freshening variability can be traced back to the signature of water in the formation regions [Church et al., 1991]. The freshening examples given above are in agreement with changes in the hydrological cycle, in which the wet (precipitation > evaporation, P>E dominance) subpolar regions have been getting wetter and *vice versa* for the dry (P<E dominance) subtropical regions  over the last 50 years

[Held and Soden, 2006; Skliris et al., 2014].

Antarctic Intermediate Water (AAIW) is characterized by a salinity minimum (core of

AAIW) and concentrated at a depth 600 -1000 m (Fig. 1), which lies within a potential density (reference to sea surface) of s0 = 27.1 -27.3 kg/m3 [
[revised manuscript text omitted]

---

## Referee Comment (RC3) · HN Waldron (Referee) · 27 Sep 2016

Pleased that I could help with improving the standard of written English. In future, I would advise you to go through this process pre-submission. The main criticism of the paper was the writing style not the science so I concentrated on this aspect. The advice you received to "tone down the conclusions" was incorporated in my edit. You had made your findings too definitive for a natural science article.

---

## Author Comment (AC1) · 27 Sep 2016

Dear Referee, We greatly appreciate you for your effort to improve this article. The second author and I are non-native speaker of English, therefor we always cannot express the meaning we have. This paper has already been rejected by DSR1, mainly due to the bad English writing. And I even cannot understand the meaning of 'tone down your conclusion' that given by those reviewers. It is quite lucky for me to get help from you. We have revised the present paper following your suggestion (Revised ms not attached). But we wonder if you have raised any scientific questions, as we have not found anyone in the paper. Express our sincere appreciation again. Best regards Wenjun Yao Ocean University of China ====================================== Wenjun Yao, Ph.D. candidate Key Lab of Physical Oceanography (Ocean University of China) Ministry

of Education 238 Songling Road, Laoshan District, Qingdao, 266100, China Email: wjimyao@gmail.com;
* * *

---

## Referee Comment (RC4) · Anonymous Referee #2 · 26 Oct 2016

**General Comments**

The main objective of this article is to report a freshening of the Antarctic Intermediate Waters (AAIW) in the South Atlantic in the past decade. The authors showed this change using a monthly climatological field based on Argo data. Two WOCE hydrographic section were used for validation. The changes in the AAIW were associated to a decrease in the Agulhas Leakage transport which in turn was related a weakening in the wind stress over the Indian Ocean. The authors did not perform any any direct calculation of freshwater transport to back up their claim.  The associations between signals and trends are done just looking at simple statistics such as mean.  The findings are interesting and maybe even important if a more careful study is carried out. For instance, show that the transport of the Agulhas Current is really changing in terms of freshwater flux. A qualitative study is not enough to be provided as evidence for the freshening.

The manuscript needs improvements in terms of clarifying some ideas but also with the written part. I find very noble that one of the reviewers is helping them to accomplish that. I have some suggestions which I included as specific suggestions below but I did not spend much time in correcting their English.  As I stated earlier, I think that as a scientific paper the manuscript needs a huge improvement specifically about their hypothesis which is that the freshening is controlled by the Agulhas Currents. They need to think more about that and include some concluding evidences.

I strongly suggest that they include more work, maybe try to follow some of their own suggestions of quantifying the different contributions to the changes.  It is my opinion that the manuscript in the present form should be rejected.

Below are some more specific comments.

**Specific Comments**

- line 33: Ocean. Bindoff ... (period instead of comma)

- line 34: Indian Ocean. Curry et al...

- line 35: "showed a" instead of "discovered the"

- Line 42: " is centered at the depths of 600 m and 1000 m

- Line 49: "The first popular one" is very informal.

- Lines 49–54 : Please, explain better what are the concepts involved in each one of the arguments.

- line 68: "greatly large" by "dominant".

- line 71: 21s?

- line 71: The acronym is not used any more.

- line 77: the uncertainty for the research has decreased or the uncertainty of the estimates?  Just decadal variability? What about the other scales?

- lines 79: Replace "discovers" by "reports".

- line 99: Replace "occupations" by "cruises". Replace elsewhere.

- Line 144: Explain why warming of surface waters lead to colder trends of AAIW.

- Line 149: "mixing with more saline surrounding waters cannot give rise to a salt loss in the salinity minimum region"

- line 152: The diagrams from JAMSTEC data is not very clear. Expand the lines in the little square.

- line 164: "P minus E"

- line 177: Replace "Here we further" by "Additionally we "

- Line 185: A transatlantic cruise is not exactly a "snapshot". Remove "snapshot".

- line 192: Replace "little lesser" by "smaller".

- line 190–192: The salinity difference is estimated between the two WOCE sections which occurred in a interval of 8 years. To compare the results from fig 3b and 5b, maybe it is a good idea to show the salinity difference divided by 8 years, so that we can have a better comparison. In any case, the difference between these two sections seems to be similar as one the one calculated as salinity trend.

- line 192–193: What is "Bellow the salinity minimum"? Refer to the neutral density values in Fig. 5b to make the discussion clearer. The comparison between Fig 3b and 5b induces to a faulty conclusion. If you show a salinity difference over a period of 8 years, then you will have 0.0013 S/year. When you compare to the trend in Fig3b, this values is not that large.

- line 194: Shouldn't be Fig. 5b?

- line 209: It is not very clear what do you call "source region". Be specific.

- line 201: Replace "supplement" by "increase".

- Line 216: reanalysis

- line 337: Fig. 6a: Interim

- Line 240: Remove "in the years 00-04

- line 243–244: Replace "concomitant with" by "in corroboration with"

- line 249–250: I don't see in the present study a concrete evidence that the AL transport has a decadal variation. The authors showed evidence for wind stress changes averaged over the Indian Ocean. How much of these changes will effectively affect the transport of water in to the South Atlantic?

- line 263–264: The evidences for such decadal changes in the South Atlantic salinity should come from strong quantitative arguments. There is no causal relation between the Atlantic and the Indian Ocean changes in the salinity. You have not actually estimated the freshwater fluxes changes from the Agulhas Current and that is the weakest point of the study. Indirect evidences are not enough.

---

## Author Response (AR1)

**General Comments**

The main objective of this article is to report a freshening of the Antarctic Intermediate Waters (AAIW) in the South Atlantic in the past decade. The authors showed this change using a monthly climatological field based on Argo data. Two WOCE hydrographic section were used for validation. The changes in the AAIW were associated to a decrease in the Agulhas Leakage transport which in turn was related a weakening in the wind stress over the Indian Ocean. The authors did not perform any any direct calculation of freshwater transport to back up their claim. The associations between signals and trends are done just looking at simple statistics such as mean. The findings are interesting and maybe even important if a more careful study is carried out. For instance, show that the transport of the Agulhas Current is really changing in terms of freshwater flux. A qualitative study is not enough to be provided as evidence for the freshening.

The manuscript needs improvements in terms of clarifying some ideas but also with the written part. I find very noble that one of the reviewers is helping them to accomplish that. I have some suggestions which I included as specific suggestions below but I did not spend much time in correcting their English. As I stated earlier, I think that as a scientific paper the manuscript needs a huge improvement specifically about their hypothesis which is that the freshening is controlled by the Agulhas Currents. They need to think more about that and include some concluding evidences.

I strongly suggest that they include more work, maybe try to follow some of their own suggestions of quantifying the different contributions to the changes. It is my opinion that the manuscript in the present form should be rejected.

Below are some more specific comments.

**Specific Comments**

• line 33: Ocean. Bindoff ... (period instead of comma)

Answer: Thanks and revised.

• line 34: Indian Ocean. Curry et al...

Answer: Thanks and revised.

• line 35: "showed a" instead of "discovered the"

Answer: Thanks and revised.

• Line 42: " is centered at the depths of 600 m and 1000 m

Answer: Thanks and revised.

• Line 49: "The first popular one" is very informal.

Answer: Replaced by 'For example, there is the …'

• Lines 49–54 : Please, explain better what are the concepts involved in each one of the arguments.

Answer: Adding 'The first standpoint states that the AAIW are primarily derived from entire subpolar sources, meanwhile the second one emphasizes the role that air-sea interaction plays in the oceans south of South America.'

• line 68: "greatly large" by "dominant".

Answer: Replaced 'greatly large' by 'substantial'.

• line 71: 21s?

Answer: Replaced by '2000s'.

• line 71: The acronym is not used any more.

Answer: The 21s? Revised as above.

• line 77: the uncertainty for the research has decreased or the uncertainty of the estimates? Just decadal variability? What about the other scales?

Answer: Revised by 'This decreases the uncertainty of estimates for the research on both seasonal and decadal variations of subsurface and intermediate waters.'

• lines 79: Replace "discovers" by "reports".

Answer: Thanks and Revised.

• line 99: Replace "occupations" by "cruises". Replace elsewhere.

Answer: Thanks and Revised.

• Line 144: Explain why warming of surface waters lead to colder trends of AAIW.

Answer: '*Church et al.* [1991] and *Bindoff and Mcdougall* [1994] has researched the counterintuitive cooling of AAIW temperature induced by warming of surface water. They showed that a warming parcel at mixed layer would subduct further equatorward, which would lead the *S-θ* curve to become cooler and fresher at a given density.'

• Line 149: "mixing with more saline surrounding waters cannot give rise to a salt loss in the salinity minimum region"

Answer: Thanks and Revised.

• line 152: The diagrams from JAMSTEC data is not very clear. Expand the lines in the little square.

Answer: Thanks and Revised. See figure below.

[Figure]

Supplementary 1

• line 164: "P minus E"

Answer: Thanks and Revised.

• line 177: Replace "Here we further" by "Additionally we "

Answer: Replaced by 'Here two …'

• Line 185: A transatlantic cruise is not exactly a "snapshot". Remove "snapshot".

Answer: Thanks and Revised.

• line 192: Replace "little lesser" by "smaller".

Answer: Thanks and Revised.

• line 190–192: The salinity difference is estimated between the two WOCE sections which occurred in an interval of 8 years. To compare the results from fig 3b and 5b, maybe it is a good idea to show the salinity difference divided by 8 years, so that we can have a better comparison. In any case, the difference between these two sections seems to be similar as the one calculated as salinity trend.

Answer: WOCE sections were observed in 2003 and 2011, while the IPRC data are from 2005 to 2014, is it proper to compare the exact salinity difference between two different periods, even though the intervals are the same.

• line 192–193: What is "Bellow the salinity minimum"? Refer to the neutral density values in Fig. 5b to make the discussion clearer. The comparison between Fig 3b and 5b induces to a faulty conclusion. If you show a salinity difference over a period of 8 years, then you will have 0.0013 S/year. When you compare to the trend in Fig3b, this values are not that large.

Answer: 'In the water layer below the salinity minimum (around $27.41\gamma^n$), the salinity increase shown in the WOCE data is relatively large (错误!未找到引用源。b).' Is it OK?

We were to show that the salinity below the salinity minimum increased for the study period, and the magnitude displayed by the WOCE was larger than that by the IPRC. Is there any contradiction ?

• line 194: Shouldn't be Fig. 5b?

Answer: Yes and Revised.

• line 209: It is not very clear what do you call "source region". Be specific.

Answer: 'The above discussion suggests that the freshening of AAIW is induced by the input of freshwater from the source regions, which are consisted of the southeast Pacific Ocean and the circumpolarly subpolar oceans (see introduction).'

• line 201: Replace "supplement" by "increase".

Answer: line210? Thanks and Revised.

• Line 216: reanalysis

Answer: Thanks and Revised.

• line 337: Fig. 6a: Interim

Answer: Thanks and Revised.

• Line 240: Remove "in the years 00-04

Answer: Replaced by 'reached a peak in the years 2000-2004'.

• line 243–244: Replace "concomitant with" by "in corroboration with"

Answer: Thanks and Revised.

• line 249–250: I don't see in the present study a concrete evidence that the AL transport has a decadal variation. The authors showed evidence for wind stress changes averaged over the Indian Ocean. How much of these changes will effectively affect the transport of water in to the South Atlantic?

• line 263–264: The evidences for such decadal changes in the South Atlantic salinity should come from strong quantitative arguments. There is no causal relation between the Atlantic and the Indian Ocean changes in the salinity. You have not actually estimated the freshwater fluxes changes from the Agulhas Current and that is the weakest point of the study. Indirect evidences are not enough.

Answer: It remains a great challenge to quantify the leakage transport, this is why we choose to qualify it in the present work.

We have added a new subsection of 4.1, and hope it can satisfy the referee's requirement.

'In the study of modeling, it is widely acceptable to quantify the leakage follows a Lagrangian approach [*Biastoch et al.*, 2009; *van Sebille et al.*, 2009]. Here a simplified strategy was employed to compute the leakage by integrating the velocity within AAIW layer (approximately between 610 and 1150m, according to 错误!未找到引用源。), which was demonstrated to result to a similar quantification to the Lagrangian one [*Le Bars et al.*, 2014]. The depth integration is along the Goodhope section (green line in 错误!未找到引用源。), using the cross-component velocity. Note that the leakage calculation is from the continent to the zero line of barotropic streamfunction, which is the separation of the Agulhas regime and the Antarctic Circumpolar Current [*Biastoch et al.*, 2015].

Before showing the transport computed from the SODA velocity data, it is necessary to verify that the SODA hydrographic data could reveal the same freshening of AAIW as the other dataset done. And in consequence, the AAIW in the South Atlantic was also shown to have freshened during period 2005-2014, though with relatively small magnitude (Supplementary 3). Yearly leakage computation within AAIW layer was employed for the period 2000-2015 (错误!未找到引用源。). It shows that the leakage in the early 2010s is smaller than that in the middle and post 2000s, forming a decreased trend in a nearly ten-year period. This estimation of leakage seems to be consistent with the below indirect estimation of AL transport.'

[Figure]

Supplementary 1

[Figure]

Fig. 6

More information please see the manuscript.

It seems that the new manuscript still has many problems, but this is the all we can do up till now.

Thanks for your comments again.

**Freshening of Antarctic Intermediate Water in the South Atlantic Ocean in 2005 - 2014**

Wenjun Yao[a,*], Jiuxin Shi[a], Xiaolong Zhao[a]

[1]Key Lab of Physical Oceanography (Ocean University of China), Ministry of Education,

Qingdao 266100, Shandong, China

*Correspondence to:* E-mail address: wjimyao@gmail.com (Wenjun Yao)

**Abstract**

Basin-scale freshening of Antarctic Intermediate Water (AAIW) is reported to have occurred in the South Atlantic Ocean during the period from 2005 to 2014, as shown by the gridded monthly means Argo (Array for Real-time Geostrophic Oceanography)

data.  This phenomenon was also revealed by two repeated  transects along a section at  30° S,  performed during the World Ocean Circulation Experiment Hydrographic Program. Freshening of the

AAIW was compensated by  a salinity increase of thermocline water, indicating a hydrological cycle intensification. This was  supported by the precipitation less evaporation change in the Southern Hemisphere from 2000 to 2014

.  Freshwater input from atmosphere to ocean surface  increased in the subpolar high-precipitation region and *vice versa* in the subtropical high-evaporation region. Against the background of hydrological cycle  changes,  a decrease in the transport of Agulhas Leakage (AL) which was revealed by the simulated velocity field, was proposed to be  a contributor  to the associated freshening of

AAIW. This  estimated variability of AL was inferred from a weakening of wind stress over the South Indian Ocean since the beginning of the 2000s, which would facilitate  freshwater input from the source region and partly contribute to the observed  freshening of AAIW.  The mechanical  analyses used in this study are  
[revised manuscript text omitted]

---

## Editor Decision (ED1)

OS-2016-54  Final comments

I believe the answers given by the authors to the latest round of comments, together with their revisions to the manuscript, are sufficient to allow publication. However, I suggest a number of changes to the wording as a way of improving the English. Many of these relate to the need for additional articles ("a" and "the"), which do not exist in Chinese. Line references are to those in the latest version of the manuscript.

Line 39 – "in agreement with enhancement of the hydrological"
Line 44 – "lies within the potential density"
Line 52 – " isopycnals" not "isopycnal"
Line 55 – "entirely" not "entire"
Line 63 – "west of the Antarctic peninsula"
Line 70 – "on the variability of AAIW"
Line 74 – "from the 1950s"
Line 82 – "reports" not "reported"
Lines 89-90 - "period 2005-2014 have been produced"
Lines 94-95 – "isopycnal surfaces, T and S profiles were first interpolated onto 1 m vertical depth intervals usinf a spline method….and a linear method"
Line 97 – "comparing them with the  other"
Line 103 – "conducted during the "
Lines 108-109 – "These two transects not only occupied almost identical station positions in the subtropical"
Lines 111-112 – "very similar to the period covered by the IPRC data (Jan 2005-Dec 2014)"
Line 125 – "of the Quick Scatterometer"
Line 126 – "and the Advanced Scatterometer"
Line 132 – "to the ocean surface"
Line 136 – "Analysis" not "analysis"
Line 138 – "provided by SODA makes it possible to evaluate"
Line 145 – delete "of" before "theta-S diagram"
Line 149 – delete "of it"
Line 150 – "at the 95% confidence level"
Line 153 – "surfaces" not "surface"
Line 158 – "parcel in the mixed layer"
Line 159 – ""decrease of the AAIW core indicates"
Line 163 – "product from JAMSTEC, are also shown"
Line 164 – "not only was the same"
Line 165 – delete "was" before "found"
Line 166 – "The isoneutral salinity increases of both IPRC and JAMSTEC"
Line 168 – put the phrase in brackets after "confidence level" on line 169
Lines 172-173 – "is estimated ….source region. This assumes that the South Atlantic"
Line 174 – "2014 per unit area"
Line 175 – delete ")" after "period."
Line 194 – "the sectional mean"
Line 200 – "has a continuous time series"

Line 204 – ""criteria and considering the number"
Line 215 – delete comma after "data"
Line 226 – "souorce regions, which consist of the southeast"
Line 232 – "In modeling studies, it is widely accepted to quantify the leakage using a Lagrangian approach"
Lines 235-236 – "which was shown to result in a similar"
Line 237 – "Good Hope" not "Goodhope"
Line 239 – "line of the barotropic"
Lines 242-244 – "hydrographic data show the same freshening of AAIW as other datasets. AAIW in the South Atlantic was also found  to have freshened"
Line 245 – ""computation within the AAIW layer was carried out for the period"
Line 247 – "middle and late 2000s, forming a decreasing trend"
Line 248 – "consistent with the indirect estimate of AL transport given below."
Line 250 – delete "above"
Line 252 – "time period, assuming that this rate"
Line 254 – delete full stop before bracket
Lines 259-260 – "This results in a salinity reduction of 0.0064, which coukd"
Lines 261-262 – "Though our estimate here is quite rough, we can state that, dsuring 2005-2014, the AL significantly influenced the salinity change"
Lines 265-267 – "An earlier study…a poleward shift of the westerly winds [Beal et al"
Lines 271-272 – "with an equatorward rather than a poleward shidt"
Line 275 – "westerly wid intensity"
Line 277 – "westerly qwind strength"
Line 280 – "from the ERA-interim wind product"
Line 282 – "WS has increased considerably"
Line 308 – "end of the time series"
Lines 314-315 – "contributions from both the source region and the AL were only quantitative"
Line 324 – " water layer is thought to be"
Line 332 – " transport is proposed"
Line 334 – "Good Hope"
Line 342 – "Our estimate further suggests that such an induced"
Line 345 – "one might ask if"

Line 347 – insert "level" before "(around 27.2…"
Line 348 – "which explains 55.4% of the variance."
Line 350 – "shows" not "displays'
Lines 352-353 – "South Indian Ocean. In addition rto these salinity changes, we also note that the salinity….Pacific was considerably less"
Line 355 – delete "at least"
Lines 358-359 – "these two contributors, and the influence they have on the world ocean circulation, through modeling studies."

---

## Author Response (AR2)

**Part 1**

Comments to the Author:

The latest review suggests that the authors have not answered all the earlier questions posed. I still have some questions of my own, so I agree with the reviewer that more work needs to be done on this paper before it can be accepted.

The authors have attempted to address at least some of the concerns of the reviewers by adding a section based on the SODA reanalysis, which also shows a decrease in salinity along the 30°S line in the South Atlantic and suggests that there has been a general decrease in the flux of water from the Agulhas since about 2004 (although the Indian-Atlantic flux anomaly was still positive during the period from 2004-2011). Thus there is a general qualitative agreement between the IPRC Argo data, the WOCE data, and the model reanalysis, but since the reanalysis is presumably based at least partly on these data it would be surprising if they did not agree. Wind data also appear to confirm the suggestion that the westerly wind strength over the Indian Ocean has decreased since about 2004.

A major question that has not been answered is whether the observed changes are caused only by changes in the Agulhas leakage. Only a few Agulhas rings cross latitude 30° S west of the 0° meridian (Duncombe Rae, 1991; Biastoch et al 2008), and yet the salinity changes in the AAIW layer are apparently seen all the way across the South Atlantic. This suggests that there should be some contribution to the freshwater flux from another source, possibly the SE Pacific, as pointed out on p.2 of the manuscript, or maybe from the Indian Ocean south of the Agulhas retroflection. How well do the estimates of Agulhas leakage from the SODA reanalysis account for the change in salinity seen in the data? Are there other data that could help determine how much of the observed changes comes from each source?

Finally, the P-E data in Fig. 4 suggest that there may be a cycle in the salinity balance with a periodicity of 30-40 years that would presumably affect the salinity along 30°S. Yet the authors do not comment on this possibility. Are there any long-term modeling studies that could support this argument, which would be additional support for their ideas?

**Answer**: Thank you for your suggestions. There are three questions asked by the editor, we will answer it one by one.

1. The authors agree that the observed changes are not only caused by the Agulhas Leakage. However, some researchers suggest that the Agulhas Leakage is the most important one among all sources of the AAIW in the South Atlantic, as stated in the manuscript: 'The influence of AL on variability of AAIW in the South Atlantic has been demonstrated to be substantial [*Hummels et al.*, 2015; *Schmidtko and Johnson*, 2012], as 50 - 60% of the Atlantic AAIW originates from the Indian Ocean [*Gordon et al.*, 1992; *McCarthy et al.*, 2012], with increased (decreased) transport of AL relating to salinification (freshening) of AAIW'.

To express the importance of AL compared to other sources, we have make some modification in the manuscript, as part of the section of Conclusions and discussions. See the manuscript: 'May someone would ask if there are any other sources that could significantly affect the AAIW in the South Atlantic Ocean, for example the Southeast Pacific (see Introduction). To clarify this question, we displayed the EOF1 pattern and its time series (called the principal components) of salinity on the $27.36\gamma^n$ (around $27.2\sigma_0$) surface (Fig. 1) in the Southern Hemisphere, which explain the largest variance of 55.4%. It shows that in 2000-2014, the most significant salinity reduction appeared in the South Indian Ocean, especially in the region of Agulhas Current System. It also displays that compared to the West Atlantic, the East Atlantic experienced the major salinity reduction, whose intermediate water is largely fed by its counterpart in the South Indian. In addition to the above salinity change distribution, we also noted that the salinity decrease in the Southeast Pacific was quite less than that in the South Indian and the South Atlantic. Therefore, it implies that the Southeast Pacific did not play an important role at least in our observed AAIW freshening.'

[Figure]

Fig. 1 (a) Pattern and (b) time series (blue: monthly, red: 13-month smooth) of EOF1 of salinity on $27.36\gamma^n$ surface. (c) Yearly mean time series of EOF1. Calculated from SODA data.

2. We have made some effort to quantify the contribution of AL to the freshening revealed by the IPRC data. See the manuscript: 'The following calculation is to simply estimate the contribution of the above AL transport change to our observed freshening. As shown by 错误!未找到引用源。, the decreased rate of AL transport could be taken to be 2 Sv in a ten-year time period. And assuming that this rate increased year by year in the study period (i.e., 0.2 Sv in the first year, 0.4 Sv in the second year, and so on.). According to *Sun and Watts* [2002], here we take the salinity difference of $\Delta S$=0.1 between the South Indian and the South Atlantic in the AAIW layers. The other parameters, including total number of seconds in a year, water thickness of the AAIW layer, the area of Region A, are taken to be $\Delta t$=365×24×3600s, $\Delta d$=500 m and $\Delta s_A$=1.09×1e13 m², respectively. Therefore, the salinity decrease from 2005 to 2014 induced by the change of AL transport, should be $(0.2+0.4+\ldots+2)\times10^6\times\Delta t\times\Delta S/(\Delta s_A\times\Delta d)$. As a result, a 0.0064 of salinity reduction was induced, which could account for approximately 53.0% of the observed freshening revealed by the IPRC data. Though our estimation here was quite roughly, through which we could be safety to state that, in the years 2005-2014, the AL behaved to significantly influence the salinity change in the South Atlantic Ocean within the AAIW layers.'

3. We haven't found any studies that describe a salinity balance with a periodicity of 30-40 years. Even the most recent study of *Schmidtko and Johnson* [2012], which discussed the hemispheric AAIW change, does not cover our study period. Note that here we only focus on the decadal variability of salinity of AAIW, we have modified the Fig. 4 in the manuscript, which would show the P-E change during 2000-2014.

[Figure]

**Part 2**

**General Comments**

In this article the authors report a freshening of the Antarctic Intermediate Waters (AAIW) in the South Atlantic in the past decade. The authors showed this change using a monthly climatological field based on Argo data and two WOCE hydrographic sections. I still have problems with salinity trends using only one decade of monthly mean fields based on Argo data, Fig. 3. Are the biennial changes significant? The Argo climatology was based on floats distribution. Specifically in the South Atlantic, the number of Argo floats is not very expressive and irregularly distributed in space and time. How would that affect the biennial mean estimation? I really urge the necessity of addressing the errors associated with the biennial means, otherwise I don't think that the differences are significant. Also, are the differences in salinity between the two WOCE sections significant? From the TS diagram, Fig. 5a, the changes in the temperature seems even more prominent. The manuscript improved considerably but still there are several mistakes and typos. The manuscript still needs some improvements to close all the weak or uncertain points. In the present form, it is my opinion that the manuscript is not ready to be published. Below are some more specific comments.

**Answer**:

First, the authors thank the referee for your valuable suggestions, and your effort to polish our manuscript. But quite unfortunately, we actually cannot answer your questions here. The official website of IPRC has not released the error estimates of their Argo gridded product. What we only can do is to compute the spatial variance of hydrographic properties in the study area. But it doesn't seem to be helpful with our research.

The significance of the salinity between the two WOCE sections was tested by using the $t$-test.

And we have not found any relevant study that discusses such data uncertainty as well. This is why we collected both the Argo gridded product of JAMSTEC and the WOCE in-situ observation data, to compare with and validate the result revealed by the IPRC data. In addition to the analysis of Agulhas Leakage and wind field in the South Indian, we have quantified the contribution of Agulhas Leakage to our observed freshening in the new manuscript. You can see the second answers in the Part 1 if you are interested in it.

**Specific Comments**

• line 32: and the period between 1985 and 1994

Answer: Two 'between' seem to be strange. The original sentence means to compare the period 1985-94 and 1960s.

• line 39: evaporation (P > E) dominance

Answer: Revised. 'in which the wet (precipitation ($P$) > evaporation ($E$), $P$ dominance) subpolar regions have been getting wetter and vice versa for the dry ($E$ dominance) subtropical'

• line 43: remove "and is"

Answer: Accepted.

• Line 44: ... surface) range of

Answer: Accepted.

• Line 65: ... the Indian Ocean transported by the Agulhas.

Answer: Accepted.

• Lines 76: Argo is not an acronym. Remove (Array for ...Oceanography). Mention as Argo profiling floats program.

Answer: We have checked it in the google scholar, the expression of "Argo (Array for Real-Time Geostrophic Oceanography)" could be found. But we have put the acronym of Argo after its meaning.

• line 82: (2005–2014) using a monthly climatology data based on Argo data.

Answer: Accepted.

• line 82: What is exactly "enhanced hydrological cycle"? Significant changes in the E-P signal?

Answer: We replaced the 'changes' in the introduction by 'enhancement' to correspond to the 'enhanced hydrological cycle' here. (line 37-41, in the new manuscript with track, 'The freshening examples given above are in agreement with the enhancement in hydrological cycle, in which the wet (precipitation > evaporation, $P>E$ dominance) subpolar regions have been getting wetter and vice versa for the dry ($P<E$ dominance) subtropical regions over the last 50 years [*Held and Soden*, 2006; *Skliris et al.*, 2014].')

• line 77: the period between 2005 and 2014 are ...

Answer: Accepted.

• lines 89: Remove "," after resolution. You should also include a reference for this dataset.

Answer: Accepted.

• line 91–94: Interporlating the T and S profiles using spline will not necessarily solve your problem of low resolution. For instance, interpolating T and S near the thermocline depth will "create" data that not necessarily fit the TS relationship in the region of study. In that sense, doing the linear interpolation sometimes is better because it will not add any new value.

Answer: Accepted. 'To reduce the error from low vertical resolution of data when computing the hydrographic values on isopycnal surface, $T$ and $S$ profiles were first interpolated onto 1 m depth intervals vertically using spline method in the intermediate water depth, and linear method in the thermocline depth.'.

• Line 92: Replace "are" by "were".

Answer: Accepted.

• Line 93: Replace "are" by "were".

Answer: Accepted.

• Line 97: Replace "are" by "were".

Answer: Accepted.

• Line 107: Don't need to repeat the same URL. Just mentioned that both are obtained in the same address.

Answer: The URL has been removed.

• line 109: November and October respectively. Remove "investigation".

Answer: Accepted.

• line 111: Replace "data" by "covered period". Replace "confirm" by "validate".

Answer: Accepted.

• line 112: You shouldn't change the dynamics of the ocean. Replace "reduce the effect of dynamic processes in the ocean" by "to smooth out some of the higher frequency variability".

Answer: Accepted.

• Line 114: Remove G from G McCarthy et al.

Answer: Accepted.

• line 119: Replace "are" by "were". Replace "display" by "investigate".

Answer: Accepted.

• line 121: Acronym should come the name.

Answer: Accepted.

• Please, examine the verb tense in the whole text. You are describing things that you have already done. The tense should be the past tense.

Answer: We have examined the full manuscript and use the proper tense of the verb.

• line 130-132: One sentence in one paragraph? Join this paragraph with the previous one.

Answer: One kind of data in one paragraph seems to be better?

• line 133: Remove "up–to–date". Also, the acronym SODA should come after its meaning. Change this order in the whole manuscript.

Answer: Accepted.

• line 154: "have" instead of "has".

Answer: Accepted.

• Line 164: "was found" instead of "found".

Answer: Accepted.

• line 167: How much is "somewhat". Put a specific value.

Answer: 'The main discrepancy between them is that the salinity reduction in the JAMSTEC data is somewhat (a mean of 0.006 between $27.1\gamma^n$ and $27.6\gamma^n$) less than IPRC and at a higher 95% confidence level.'

• Line 168–170: Explain clearly how did you came up with a 15 mm/y, a one dimensional estimate for the whole ocean.

Answer: 'The freshwater gain for the basin-scale salinity decrease of AAIW (mean salinity difference of 0.012 between $27.1\gamma^n$ and $27.6\gamma^n$ over a mean water mass thickness of 500 m) was estimated at 17mm $yr^{-1}$ in its source region (Assuming the case that the South Atlantic only experienced freshwater input and nothing changed, thus the relationship between the salinity in 2005 and 2014 in unit area was roughly $S_{2005}*500 = S_{2014}*(500+\Delta d)$. Here $S_{2005} = S_{2014}+0.012$ and $\Delta d$ is the freshwater gain during the covered period).' And we found that 17 mm/y was better.

• line 179–185: First of all, the P-E change in 2005 seems to be about 0.01 mm/d and not 0.2 mm/d.

Answer: We had made some mistakes. The unit in Fig. 4b is mm$\times10^{-1}$/d, thus the increased annual freshwater input was 0.02 mm $day^{-1}$.

• line 189: The $\theta$S diagram could also imply that the temperature incresed from 2003 to 2011.

Answer: For example, along the 27.3 neutral density surface, the point of red line (represent 2011) lies left to and lower than that of black line (2003), which means that the salinity reduces and temperature decreases.

• line 209: "give" instead of "giver".

Answer: The right one is 'given'.

• lines 215–223: The authors didn't show that all the changes in the AAIW is solely due to the Agulhas contribution. The difference in salinity (fig 5b) comes from a wide range of density and could come from other sources. That discussion is just a speculation.

Answer: Please see the first and second answers in the Part 1.

References

[revised manuscript text omitted]

---

## Author Response (AR3)

Dear Editor:

Thank you very much for your effort to polish our paper, and we have accepted it to revise our manuscript.

Some important revisions we would like to inform you include:

1. The author affiliation of the first and second authors have been changed a little.

2. We have added the third author, Xiaolong Zhao, who provided us with the technique of calculation of barotropic streamfunction. Actually we have changed it in our first major revised manuscript.

If you need the latest manuscript with editing line or there is any problem, please inform me.

Best regrads.

Wenjun Yao.

Ocean University of China.

2017/6/2